# Mechanisms of Action of Autophagy Modulators Dissected by Quantitative Systems Pharmacology Analysis

**DOI:** 10.3390/ijms21082855

**Published:** 2020-04-19

**Authors:** Qingya Shi, Fen Pei, Gary A. Silverman, Stephen C. Pak, David H. Perlmutter, Bing Liu, Ivet Bahar

**Affiliations:** 1Department of Computational and Systems Biology, School of Medicine, University of Pittsburgh, Pittsburgh, PA 15213, USA; qingya@pitt.edu (Q.S.); fep7@pitt.edu (F.P.); 2School of Medicine, Tsinghua University, Beijing 100084, China; 3Department of Pediatrics, School of Medicine, Washington University in St. Louis, St. Louis, MO 63130, USA; gsilverman@wustl.edu (G.A.S.); stephen.pak@wustl.edu (S.C.P.); perlmutterd@wustl.edu (D.H.P.)

**Keywords:** autophagy, quantitative systems pharmacology, signal transduction, machine learning, drug-target interactions, mechanism of action, mTOR, PKA, PI3K

## Abstract

Autophagy plays an essential role in cell survival/death and functioning. Modulation of autophagy has been recognized as a promising therapeutic strategy against diseases/disorders associated with uncontrolled growth or accumulation of biomolecular aggregates, organelles, or cells including those caused by cancer, aging, neurodegeneration, and liver diseases such as α1-antitrypsin deficiency. Numerous pharmacological agents that enhance or suppress autophagy have been discovered. However, their molecular mechanisms of action are far from clear. Here, we collected a set of 225 autophagy modulators and carried out a comprehensive quantitative systems pharmacology (QSP) analysis of their targets using both existing databases and predictions made by our machine learning algorithm. Autophagy modulators include several highly promiscuous drugs (e.g., artenimol and olanzapine acting as activators, fostamatinib as an inhibitor, or melatonin as a dual-modulator) as well as selected drugs that uniquely target specific proteins (~30% of modulators). They are mediated by three layers of regulation: (i) pathways involving core autophagy-related (ATG) proteins such as mTOR, AKT, and AMPK; (ii) upstream signaling events that regulate the activity of ATG pathways such as calcium-, cAMP-, and MAPK-signaling pathways; and (iii) transcription factors regulating the expression of ATG proteins such as TFEB, TFE3, HIF-1, FoxO, and NF-κB. Our results suggest that PKA serves as a linker, bridging various signal transduction events and autophagy. These new insights contribute to a better assessment of the mechanism of action of autophagy modulators as well as their side effects, development of novel polypharmacological strategies, and identification of drug repurposing opportunities.

## 1. Introduction

Autophagy is the process that enables self-clearance of potentially damaging protein aggregates, dysfunctional organelles (e.g., mitochondria by mitophagy), or entire cells, and as such it is one of the most significant regulatory processes that ensures healthy functioning of cells [1,2]. During starvation, autophagy can serve as an essential source of energy for survival, and in general it plays a housekeeping role to maintain homeostasis and viability of cells [3]. In addition, autophagy has been shown to be instrumental in erasing pathological responses or maintaining physiological functions of the cells such as anti-aging, tumor suppression, and antigen presentation [4]. 

Once autophagy is activated, e.g., by stress stimuli such as starvation, a cascade of reactions are triggered that lead to the formation of a double-membrane-bound vesicle called an autophagosome [5], which encapsulates the cytoplasmic aggregates or dysfunctional organelles. The subsequent fusion of the autophagosome with the lysosome to form the autolysosome permits the lysosomal enzymes to digest and degrade the contents of the autophagosome [5,6]. 

According to our current understanding, autophagic responses are modulated by many events, in which the kinase mammalian target of rapamycin (mTOR) plays a major role as a master regulator of autophagy [7]. mTOR, a serine/threonine kinase, is a downstream regulator of the phosphoinositide 3-kinase (PI3K) pathway. It suppresses autophagy induction by inhibiting the activation of the UNC-51-like autophagy activating kinase (ULK) complex [8]. The complex of activated ULK1 with autophagy related 1 (Atg1) protein, Atg1-ULK, would otherwise receive signals on cellular nutrient status and recruit downstream autophagy-related (ATG) proteins to the autophagosome formation site, thus initiating an autophagic response [9]. mTOR further acts as a negative regulator of autophagy by preventing the general expression of lysosomal and autophagy genes through phosphorylating transcription factor EB (TFEB) [7,8]. 

Other important regulators of autophagy include cAMP-dependent protein kinase A (PKA) [10] and other proteins involved in Ca^2+^ signaling pathways. Ca^2+^ ions have complex effects [11,12]: Cytoplasmic Ca^2+^ activates Ca^2+^/calmodulin-dependent kinase kinase β (CaMKKβ), which activates AMP-activated protein kinase (AMPK) to promote autophagy [13]. However, CaMKKβ also activates calpain, which inhibits autophagy by cleaving autophagy related 5 (Atg5) protein [14]. 

In the last few decades, there has been a surge in the number of autophagy-modulating drugs targeting these well-known autophagy regulators. For example, rapamycin that inhibits mTOR is known as one of the most important autophagy activators [15], and wortmannin which inhibits PI3K is known as a significant autophagy inhibitor [16,17]. In the meantime, autophagy modulation has been receiving increased attention as a strategy for developing therapeutics [18]. Autophagy has been widely accepted as cytoprotective against neurodegenerative diseases, and many clinical interventions are moving forward to increase autophagy as a therapeutic intervention [19]. For example, rapamycin is regarded as an important pharmacological agent against Alzheimer’s disease (AD) [20]. Various autophagy-enhancing drugs (such as carbamazepine [21], fluphenazine [22], and glibenclamide [23]) have been identified to have therapeutic effects against α1 antitrypsin deficiency (ATD) caused by the aggregation of the misfolded α1-antitrypsin Z (ATZ) proteins in the liver. Besides, autophagy dysfunction is also highly associated with cancer, and many autophagy-targeting drugs also serve as anticancer drugs [24]. For instance, metformin, an autophagy enhancer, is a drug against myeloma [1].

In this study, we carried out a comprehensive analysis of a set of 225 known autophagy modulators selected from DrugBank v5.1.1 [25] (Appendix A), their 993 target proteins, and 294 associated pathways in order to obtain a systems-level understanding of the common mechanisms of autophagy modulation, as well as specific mechanisms of selected modulators. The set of modulators consists of 174 activators, 31 inhibitors, and 20 compounds acting as dual-modulators. We identified 1831 known interactions between these drugs and their targets, and predicted 368 novel interactions using our probabilistic matrix factorization (PMF)-based application programming interface [26,27,28]. Of these predictions, 75 were consistent with recently published experimental data (not yet deposited in DrugBank [25]). 

Our results show that autophagy modulators are structurally diverse and functionally promiscuous. Their target proteins are commonly expressed in the brain and liver tissues and mainly function as tyrosine kinases, calcium channels, aminergic GPCRs, or nuclear hormone receptors. The most frequently targeted proteins do not directly take part in the autophagy pathway, but indirectly regulate the activity/expression level of core ATG proteins through differential signaling pathways. Through gene ontology (GO) annotation and pathway enrichment analysis, we distinguished the most important signaling pathways that regulate autophagy. Our results strongly point to the crucial role of PKA as a hub conveying autophagy signals, and directly regulating ATG proteins and their gene expression. 

## 2. Results

### 2.1. Autophagy Modulators Are Structurally Diverse and Have Promiscuous Functions

We manually curated the role of the selected 225 autophagy modulators based on data/reports in the literature, and we classified them into three functional categories: activators (*n* = 174), inhibitors (*n* = 31), and dual-modulators (*n* = 20). Here, dual-modulators refer to drugs that can both positively and negatively regulate autophagy depending on the biological context. For instance, melatonin is a dual-modulator because it reduces autophagic activity in tumor trophoblast cells while it enhances it in normal cells [29]. 

As a first step, we examined to what extent the autophagy modulators within each group share structural and functional similarities. Appendix A shows that the selected autophagy modulators are highly diverse. Within all three categories, most of the drug pairs have low structural and functional similarities (<0.5) evaluated based on their 2D fingerprints and drug–target interaction patterns extracted from DrugBank, respectively. However, we can distinguish clusters of drugs that have similar structures, which also share similar interaction patterns with targets. For example, rapamycin and its derivatives everolimus, temsirolimus, and ridaforolimus (Appendix A, panels a and b, activators #28-30, enclosed in light yellow ellipse) activate autophagy by inhibiting mTOR [30], and retain the same structure, presumably targeting the same site on mTOR. In contrast, the upper right white region in panel b (drugs #141-166, enclosed in pink ellipse) indicates several autophagy activators (e.g., fluphenazine, pimozide, clonidine, paroxetine, triflupromazine, chlorpromazine, citalopram, nortriptyline, fluspirilene, doxazosin, amiodarone, flunarizine, verapamil, and dronedarone) that share similar interaction patterns. Most of them are therapeutic agents for mental disorders such as schizophrenia, depression, and anxiety disorders, or for cardiovascular diseases such as high blood pressure, angina, and arrhythmia, regulating autophagy through cAMP- and Ca2+-signaling pathways. However, panel a indicates that they are structurally heterogeneous, suggesting that they have different targets on their shared pathways. 

Further examination of Appendix A shows that the macrolide antibiotics erythromycin, clarithromycin, and azithromycin (inhibitors #5-7 in panels c and d) inhibit autophagy via endoplasmic reticulum (ER) stress-mediated C/EBP homologous protein (CHOP) induction [31], but despite their close structural similarity (see enclosed region in panel c), they exhibit distinctive interaction patterns with their targets (panel d), suggesting different mechanisms of action. The phenothiazine antipsychotics thioridazine and trifluoperazine (panels e and f, dual-modulators #2-3) regulate autophagy through dopamine receptors [32] and share close structural similarities.

### 2.2. Selected Autophagy Modulators Are Distinguished by Their High Promiscuity

The space of proteins targeted by autophagy modulators is quite broad. We identified 993 such proteins (Appendix A) composed of 706 targets associated with 174 activators, 374 associated with the 31 inhibitors, and 94 targets associated with the 20 dual-modulators, which show partial overlaps, as presented by the Venn diagram in Figure 1a (right). The space of drugs and targets may be viewed as a bipartite network, with multiple connections (drug–target interactions). The number of targets connected to a given modulator will be referred to as the degree of the modulator. The higher the degree, the more promiscuous the modulator. 

Based on DrugBank v5.1.1, we identified a total of 1831 types of known drug–target interactions between these autophagy modulators and their target proteins. Specifically, the activators, inhibitors, and dual-modulators were involved in 1339, 389, and 103 known interactions with their targets (Figure 1a).

In principle, if each autophagy modulator had one distinctive target, we would have 225 targets (or even less as some targets could overlap), but this is not the case: Several modulators are highly promiscuous (Appendix A) in the sense that they interact with multiple targets. The detailed distribution of the multiplicity of targets among modulators is summarized in Figure 1b. Only 25%–32% of the modulators interact with a single drug (red portions of the bars), depending on the type of modulator. It is interesting to note that almost 20% of activators have more than 10 targets (yellow portion at the top of the leftmost bar), pointing to the high promiscuity of ~30 activators. Specifically, 31 out of 174 (17.8%) activators, 3 out of 31 (9.7%) inhibitors, and 3 out of 20 (15.0%) dual-modulators have at least 10 known targets.

Figure 2 provides more details on the identity of specific modulators with high promiscuity. The ordinates show the number of targets for autophagy activators (panel a), and all autophagy inhibitors (panel b) and dual-modulators (panel c), and the abscissa shows the modulators reordered in decreasing promiscuity. See also Appendix A for the target number distribution of additional (less promiscuous) activators. We observed an average of 7.7 interactions per activator, 2.9 per inhibitor (if we exclude one outlier, fostamatinib; see below), and 5.3 per dual-modulator. The insets in the three panels of Figure 2 show the histograms of modulators with different promiscuity. 

Notably, of the 1339 interactions in which activators are involved, 145 are made by the most promiscuous activator, copper (Figure 2a, *leftmost* bar), followed by zinc (124 interactions/targets), artenimol (78 targets), and olanzapine (48 targets). Transition metals copper and zinc are non-structural intracellular signaling mediators and essential elements in many enzymes. They both can induce autophagy by activating kinases such as mitogen-activated protein kinase (MAPK) [33,34]. A recent study shows that copper can even directly bind to ULK1/2 to enhance its pro-autophagic activity [35]. The antimalarial drug artenimol is a derivative of artemisinin that effectively kills *P. falciparum* by producing reactive oxygen [36] and targeting multiple biological processes otherwise essential for parasite survival [37]. The oxidative stresses induced by artemisinin and its derivatives also stimulate autophagy in the host cells [38,39]. As an antipsychotic drug, olanzapine mainly targets receptors of neurotransmitters such as dopamine, serotonin (5-HT), and γ-aminobutyric acid (GABA), which have been reported to regulate autophagy [32,40,41]. Further, olanzapine also induces autophagy by activating AMP-activated protein kinase (AMPK) signaling [42,43] and upregulating the expression of the products of ATG genes such as Atg5 and Beclin-1 [44]. 

The most promiscuous autophagy inhibitor is fostamatinib (303 targets). Fostamatinib is an FDA-approved drug for the treatment of chronic immune thrombocytopenia. It has been reported that fostamatinib suppresses autophagy by inhibiting the spleen tyrosine kinase (STK) [45]. Fostamatinib also inhibits critical autophagy regulators including ULK1/2, serine/threonine-protein kinase (TBK1) [46], death-associated protein kinase 1 and 2 (DAPK1/2) [47,48], and leucine-rich repeat serine/threonine-protein kinase 2 (LRRK2) [49]. 

The most promiscuous dual-modulator of autophagy is calcium (20 targets), followed by melatonin (10 targets). Ca^2+^ is a cofactor of many enzymes and also acts as a second messenger mediating various signal transduction pathways. It targets calcium channels (e.g., Ca^2+^ voltage-dependent channel subunit α1c, CACNA1C), calcium-transporting ATPase (e.g., ATP2C1), and downstream calcium-binding proteins (e.g., calmodulin). Cytoplasmic Ca^2+^ inhibits autophagy via activation of calpain and the inositol trisphosphate receptor (IP_3_R), while it also enhances autophagy via activation of AMPK [11]. Our previous study suggested that Ca^2+^/calmodulin-dependent kinase kinase β (CaMKKβ) plays a key role in regulating the balance between these opposing actions and acts as a determinant of the effect of competing roles of cytoplasmic Ca^2+^ in autophagy regulation [11].

To further identify potential drug-target interactions in DrugBank, we used our PMF model (see Materials and Methods) for predicting new targets for autophagy modulators. This analysis led to 368 novel interactions with high confidence scores (>0.6), which included 12 additional targets. These pairs (not reported in the DrugBank) are listed in the Appendix A. The number of predicted targets for each drug are shown in Figure 2 by the *light*-shaded portions of the bars. Notably, 75 (out of 368) predicted pairs are consistent with already published experimental data [50,51,52]. Table 1 lists 20 of them, along with the corresponding experimentally measured binding affinities, and others can be seen from the entries in Appendix A where experimental binding affinities are listed. For example, the promiscuous drugs olanzapine and fostamatinib mentioned earlier were found to bind to serotonin receptor 1F (HTR1F) and fibroblast growth factor receptor 4 (FGFR4), respectively. Fluphenazine, fluspirilene, thioridazine, sertindole, and trifluoperazine were identified to bind to dopamine receptor D3 (DRD3), and experiments indicate their high affinities (*K_i_* < 5 nM). We also detected various drug–target interactions with modest-to-weak affinities (e.g., the interaction between imatinib and Fms related receptor tyrosine kinase 3 (FLT3)).

Taken together, the 225 autophagy modulators we selected are diverse in terms of both structure and function. Several autophagy modulators, and especially autophagy activators, are distinguished by their highly promiscuous effects. The additional drug–target interactions we identified using the PMF model are plausible, thus complementing existing knowledge and also helping generate new hypotheses that await further testing. To have a better understanding of the effects of autophagy modulators, we next analyzed their target proteins.

### 2.3. Frequent Targets of Autophagy Modulators Are Not ATG Proteins But Their Regulators

As described above, the set of autophagy modulators have 1005 targets (993 known/reported in DrugBank plus 12 identified here). While drugs were highly promiscuous, the targets usually had a smaller number of connections to drugs in the bipartite network, representing drug–target pairs. We observed an average of 1.9 interactions per target, and 66.9% of the targets interacted with one single drug. 

We show in Figure 3a the most frequently targeted proteins (51 of them), which interact with five or more known autophagy modulators. The bars display the corresponding numbers of modulators broken down by color-coded categories, with *dark* and *light* shaded portions referring to known and predicted interactions, respectively. We refer to this set of proteins as frequent targets (FTs) of autophagy modulators. The distribution of autophagy targets over protein families, extracted from family memberships given in Uniprot [53] is presented in Figure 3b. Kinases (32.17%), ion channels (6.76%, including ligand-gated ion channels, Ca^2+^ channels, and Na^+^ channels), and G protein coupled receptors (GPCRs) (5.85%) are three major protein families targeted by our autophagy modulators. However, this presumably reflects the historical dominance of these families among drug targets, noted in earlier studies [54]. More meaningful are the identities of the specific targets belonging to these families, which are analyzed next in relation to ATG proteins.

In order to see whether FTs overlap with the targets involved in autophagy pathways, we compared them to the 128 proteins participating in the canonical autophagy pathway (KEGG id: hsa04140; in the following, we refer to these 128 proteins as ATG proteins). Thirty-two out of 1005 targets of autophagy modulators were found to be ATG proteins, including mTOR, ULK1/2, AMPK, CAMKKβ, insulin (INS), AKT, PI3K, PKA, PKC, phosphoinositide-dependent kinase 1 (PDK1), Raf1, ERK1/2, JNK2/3, Beclin-1, BCL2, DAPK1/2/3, cathepsin L (CTSL), inositol-requiring enzyme 1 (IRE1), eukaryotic translation inhibition factor 2α kinase 4 (EIF2AK4), and hypoxia inducible factor 1α (HIF-1α) (see the rows highlighted in *green* in Appendix A). Among them, 15 are targeted by autophagy activators, 26 by inhibitors, and two by dual-modulators. Among the ATG proteins, only mTOR is an FT. This analysis therefore showed that the majority of ATG proteins are not FTs of autophagy modulators. 

The question was then to assess what the FTs are, and why they are the targets of many autophagy modulators. We next conducted a more detailed analysis of the FTs. 

Figure 3a shows that a nuclear hormone receptor (NHR) (subfamily 1 group I member 2 (NR1I2) is the most frequently targeted protein. NR1I2 is found here to interact with 14 activators, one inhibitor, and one dual-modulator. The NR1I2 gene encodes the pregnane X receptor that regulates the transcription of many genes involved in the metabolism and secretion of potentially harmful endogenous and xenobiotic compounds, and in the homeostasis of glucose, lipid, cholesterol, bile acid, and bilirubin. The current analysis strongly suggests that it also activates the genes involved in autophagic elimination. There are six more NHRs that are FTs: estrogen receptor 1 and 2 (ESR1, ESR2), androgen receptor (ANDR), and peroxisome proliferator activated receptors γ, α, and δ (PPARG, PPARA, PPARD). Numerous studies have reported that these transcription factors regulate the expression of core ATG proteins. Specifically, ESR1 and ESR2 regulate mTOR expression (see review [55]). ANDR regulates the transcription of ULK1/2, Atg4B/D, and transcription factor EB (TFEB), a master regulator of autophagy genes [56]. PPARs are known to regulate autophagy via AMPK, phosphatase, and tensin homolog (PTEN) expression [57]. 

Eleven FTs directly participate in calcium signaling, including calmodulin and 10 ion channel family members: Ca^2+^ voltage-gated channel subunit α1c (CACNA1C) and other members of the same family (CACNA1D, CACNB2, CACNA1S, CACNA1H, CACNA2D1, CACNB1, CACNA2D2, K^+^ voltage-gated channel subfamily H member 2 (KCNH2), and Na^+^ voltage-gated channel α subunit 7 (SCN6A). We note that calmodulin interacts with 9 activators and 3 dual-modulators, consistent with the complex role of cytoplasmic Ca^2+^ in regulating autophagy.

Eighteen FTs are GPCRs, consisting of α1A adrenergic receptor ADRA1A and its family members (ADRA1B, ADRA1D, ADRB1, ADRB2, ADRA2A, ADRA2B, ADRA2C), dopamine receptors (DRD1, DRD2), histamine receptor HRH1, serotonin receptors (HTR2A, HTR2C), and muscarinic receptor CHRM1-5. GPCRs regulate autophagy through diverse downstream signaling pathways [58]. For example, ADRA1 activation induces protein kinase B (AKT)-mediated autophagy. β-adrenergic receptors (ADRB1 and ADRB2) promote autophagy via the cAMP pathway [59]. CHRM3 regulates autophagy via the CaMKKβ-AMPK pathway [60]. DRD1 stimulation induces autophagy via a cAMP-dependent but EPAC-independent mechanism [61], while the D2-like family of dopamine receptors (DRD2-4) positively regulates autophagy through AKT-mTOR and AMPK pathways [51,62]. Furthermore, activation of HTR2 inhibits autophagy via mTOR-independent p70S6K and 4E-EP1 phosphorylation [63].

As shown in Figure 3b, 32.17% of the targets are protein kinases mediating intracellular signal transduction. Among them, seven are FTs, including mTOR and several receptor tyrosine kinases (RTKs): epidermal growth factor receptor (EGFR), vascular endothelial growth factor receptor 2 (KDR), Abelson murine leukemia viral oncogene homolog 1 (ABL1), proto-oncogene c-KIT (KIT), and platelet-derived growth factor receptors A and B (PDGFRA and PDGFRB). Accumulating evidence suggests an interplay between RTK signaling and autophagy [64]. In particular, EGFR activation has been shown to inhibit autophagy via Beclin-1 phosphorylation [65]; KDR activation suppresses autophagy by upregulating FoxO1 expression; and ABL1 regulates early and late stages of autophagy by phosphorylating Beclin-1 and promoting lysosomal trafficking, respectively [66]. Furthermore, mutant KIT in acute myeloid leukemia (AML) cells triggers autophagy via STAT3 activation. Inhibition of PDGFRs can induce autophagy through the PI3K-AKT pathway [67].

This detailed analysis points to several targets whose pro- or anti-autophagic roles are consistent with experimental data. It also shows that the ATG proteins, except for mTOR, are not necessarily the direct targets of autophagy modulators, and a broad number of activators, inhibitors, and dual-modulators instead target proteins involved in the transcription regulation of genes involved in autophagy, or in intracellular signal transduction mainly via protein kinases and Ca^2+^ and Na^+^ channels. Notably, six targets interact with all three categories of autophagy modulators (see the Venn diagram in Figure 1 (*right*)): two nuclear hormone receptors (NR1I2 and ESR1), two proteins involved in potassium channeling and transport (KCNH2 and ATP4A, respectively), a GPCR (ADORA2A), and the casein kinase II subunit α (CSNK2A1, also known as CK2α) (highlighted in *yellow* in Appendix A).

### 2.4. Autophagy Modulation Targets Are Highly Expressed in the Liver and in the Brain 

Tissue-based enrichment analysis using the ARCHS^4^ database [68] suggests that autophagy modulation targets are highly expressed in the liver and in the brain (including various brain regions: cingulate gyrus, superior frontal gyrus, prefrontal cortex, dorsal striatum, and midbrain) (Figure 3c). This could be attributed to the occurrence of aggregation-prone diseases in both organs, e.g., ATZ aggregation in the liver, and many neurodegenerative disorders in the brain. Compared to other tissues, 165 targets were found to have elevated expression levels in the brain using the Human Protein Atlas [69] (Appendix A). 

Overall, our results in the above two subsections show that target proteins of autophagy modulators are commonly expressed in the brain and liver tissues and mainly function as tyrosine kinases, calcium channels, aminergic GPCRs, or nuclear hormone receptors. Few frequent targets directly take part in the canonical autophagy pathway, but many of them regulate the expression and activity of core ATG proteins through transcriptional regulation and signaling pathways. We next carried out an in-depth analysis of these pathways to understand the pharmacological mechanisms that underly autophagy modulation.

### 2.5. Functional Analysis of the Targets Reveals Enriched Pathways Implicated in Autophagy Modulation 

Figure 4a shows the gene ontology (GO) annotation terms enriched in targets of autophagy modulators. The top 20 GO cellular components include the nucleus, extrinsic components of the cytoplasmic side of the plasma membrane, and voltage-gated calcium/sodium channel complex. The top 20 GO molecular functions include various protein kinase activity-related and ion channel-related terms. The top 20 biological processes comprise a variety of protein phosphorylation-related and signal transduction-related processes (see Appendix A for an extended list). These results are consistent with our observations in Section 2.3 and highlight the importance of RTK signaling, Ca^2+^ signaling, and downstream gene transcription regulation. To quantitatively assess the major pathways affected by autophagy modulators, we carried our pathway enrichment analysis using the KEGG database [70].

Our analysis indicated that the known and predicted targets of autophagy modulators take part in 294 pathways (Appendix A). Figure 4b highlights the top 20 enriched pathways in three distinct KEGG groups: signal transduction, cellular processes, and organismal systems. The enriched pathways in other groups (human diseases, metabolism, and genetic information processing) can be found in Appendix A. We found that the signal transduction group contains three tiers of autophagy regulation: (1) pathways that directly involve core ATG proteins: mTOR-, PI3K-AKT-, and AMPK-pathways; (2) upstream signaling pathways that regulate the activity of ATG proteins: Ca^2+^-, cAMP-, and RTK/Rap1/Ras/MAPK-pathways; and (3) transcription pathways that regulate the expression of ATG proteins, mainly HIF-1-, NF-κB-, and FoxO-pathways and their regulators such as nuclear hormone receptors NR1I2 and ESR1. 

Though the enrichment scores of pathways in the cellular processes group are relatively low compared to those in the signal transduction group (due to the larger number of proteins involved in the cellular processes pathways), three autophagy pathways in KEGG were captured by our analysis, including autophagy-animal, autophagy-others, and mitophagy. Various programmed cell death pathways also exhibit relatively high scores, including apoptosis, necroptosis, and ferroptosis. This is due to a complex crosstalk between autophagy and programmed cell death pathways [71,72,73], which has been shown to process cellular stress signals and confer cell-fate decisions [11]. Interestingly, five cell adhesion pathways are top-ranked among cellular processes: regulation of actin cytoskeleton, focal adhesion, gap junction, adherens junction, and tight junction. These pathways regulate the cytoskeletal dynamics, which play a vital role in the biogenesis of autophagosomes from the ER membrane as well as their translocation and fusion with the lysosomes [74]. Other enriched pathways in the organismal systems class relate to the physiological functions of autophagy, such as neuron survival/functioning [75], insulin production/sensitivity [76], circadian rhythm [77], and longevity [78]. 

We further analyzed the distribution of pathways for each category of autophagy modulators using QuartataWeb [28]. Appendix A presents the list of pathways with high enrichment scores (*p* value < 0.05), including 63 targeted by activators, 24 targeted by inhibitors, and 106 targeted by dual-modulators. These pathways were further classified into finer subclasses of KEGG pathways. The subclasses of pathways enriched in targets of autophagy activators, inhibitors, and dual-modulators are shown in Figure 4c. Signal transduction pathways dominate all three categories, along with the endocrine system, a subclass of the organismal systems. Among the top-ranked endocrine system pathways, we distinguish the estrogen signaling pathway targeted by 131 activators, 25 inhibitors, and 14 dual-modulators; whereas melanogenesis is exclusively targeted by autophagy activators (see Appendix A). Recent studies show that rottlerin, an autophagy activator, regulates melanogenesis by targeting the cAMP-CREB signaling pathway [79]. 

Cancer pathways are also commonly targeted by autophagy modulators (Figure 4c). This is particularly clear in the case of inhibitors. Our results reflect the fact that many autophagy modulators are anti-cancer drugs, which trigger programed cell death and introduce cellular stresses. In response to such stresses, cancer cells may trigger autophagy at the same time to protect themselves. Autophagy is a first rescue mechanism for cells before proceeding to cell death in the presence of stronger insults [11], and its interference may thus delay, if not prevent, tumor cell death. Thus, autophagy inhibition has been identified as a strategy of cancer therapeutics [80]. In particular, the breast cancer pathway is highly enriched in targets of autophagy inhibitors (*p* value = 1.95 × 10^−5^) but not activators or dual-modulators. This is consistent with the observation that autophagy inhibitors such as SB02024 increase the sensitivity of breast cancer cells to chemotherapy [81]. 

We also noticed that autophagy inhibitors target more immune system pathways and infection diseases than activators and dual-modulators (respective *green* and *cyan* portions of the bars in Figure 4c). Immune pathways specific to autophagy inhibition include Fc receptor (FcR) signaling, Toll-like receptor (TLR) signaling, T cell receptor (TCR) signaling, c-type lectin receptor (CLR) signaling, and chemokine signaling pathways (Appendix A). It has been reported that stimulation of TLRs, FcRs, TCRs, and CLRs can induce autophagy [82,83]. We thus hypothesize that the mechanism of action of selected autophagy inhibitors is through blockage of these pathways.

Taken together, our enrichment analysis reveals that autophagy activators, inhibitors, and dual-modulators usually affect the same group of pathways, although a few selected pathways seem to be specifically targeted by autophagy activators (e.g., melanogenesis pathway) or inhibitors (e.g., TLR pathway). Signal transduction mechanisms shared by the three categories of autophagy modulators include those regulating the activity of ATG proteins directly (e.g., AKT pathway) or indirectly (e.g., calcium pathway), in addition to those involved in their transcriptional regulation (e.g., NF-κB and NHR pathways). We next investigated how different signal transduction pathways interconnect to control/regulate autophagy.

### 2.6. PKA, PI3K, AKT, and mTOR Play a Central Signal Transduction Role in Mediating Autophagy

We identified seven signal transduction pathways highly enriched in targets of all three categories of autophagy modulators: cAMP-, cGMP-PKG-, calcium-, Rap1, Ras, MAPK-, and PI3K-AKT-signaling pathways, schematically shown in Figure 5a. These pathways each contain a large number (between 43 and 108) of targets of autophagy modulators and they were each targeted by a large number (39 to 73) of modulators (see the rows colored *light blue* in Appendix A).

Next, we analyzed the known and predicted targets of autophagy modulators and identified those shared by one or more of these pathways as well as the canonical autophagy pathway from KEGG. This led to 15 ATG-signaling targets. These target proteins, distinguished by their enhanced propensity to be involved in ATG signal transduction, are PKA, PI3K, AKT, MEK1/2, Raf1, JNK1, ERK, Bad, INS, IGFR, mTOR, PP2A, Bcl-2, PDK1, AMPK, and TBK1. Among them, PKA and PI3K each take part in four of the seven signal transduction pathways (Figure 5b), and AKT in six pathways, and as such they play a key role in mediating autophagic responses. A detailed mapping of these targets to signal transduction pathways is presented in Figure 5b. Three color-coded entries therein indicate whether the ATG-signaling target is associated with an autophagy activator (*green*), inhibitor (*red*), and/or dual-modulator (*blue*). We note in Figure 5b (and in Appendix A) that PKA, PI3K, and mTOR are targeted by all three categories of autophagy modulators, further supporting the key role of PKA and PI3K, and highlighting that of mTOR as well. The diagram also shows that AKT, MEK1/2, Raf1, and ERK take part in at least five pathways. Interestingly, the MAPK pathway components MEK1/2, Raf1 and JNK1 are targeted by both activators and dual-modulators but not inhibitors. In contrast, AKT and IGFR are targeted by activators and inhibitors, but not dual-modulators. 

Overall, this analysis draws attention to the central role of PKA as a mediator of ATG signal transduction events. PKA regulates three key players, mTOR, MEK, and Raf1, which in turn regulate autophagy (Figure 5a). We also note the critical role of PI3K that regulates AKT, which in turn regulates mTOR. We next constructed a comprehensive model of an ATG signal transduction network that provides an integrated view of these complex interactions.

### 2.7. An Integrative Roadmap for Autophagy Modulation 

We constructed a unified signaling network that integrates the above described pathways, which allows us to visualize how targets of autophagy modulators regulate autophagy (see Figure 6a). Core ATG proteins are colored *yellow*; the associated transcription factors are in *light blue*. The above described major ATG-signal transduction pathways are displayed in distinct colors as described in the caption, and the crosstalk between these pathways are schematically shown in Figure 6b. We also indicate in *gray* boxes the six proteins that are targeted by all three categories of autophagy modulators. This comprehensive network of ATG-signal transduction pathways further highlights the role of PKA (at the center) as a key mediator of the effects of autophagy modulators, bridging various signal transduction events to autophagy and regulating the activity and gene expression of ATG proteins. *Yellow* ellipses highlight key ATG-signal transduction proteins. 

This model provides a comprehensive framework for quantitative analysis of ATG-signal transduction pathways and clears the way to building polypharmacological treatment hypotheses. Figure 6a indicates several autophagy modulators that target different components, including those binding to upstream channels/receptors such as calcium or potassium channels (*top left*), the GPCR ADORA2A (*top right*), the downstream key signal transduction elements such as PI3K, AKT, and mTORC1, or transcription factors such as HIF-1α.

## 3. Discussion

Pharmacological modulation of autophagy has great therapeutic potential and attracted a great deal of attention in the past two decades [18]. A variety of autophagy-focused databases have been developed to provide annotation information for autophagy-related proteins (such as the Autophagy Database [84]), the interactions of autophagy components (e.g., Autophagy Regulatory Network [85], THANATOS [86]), and autophagy-modulating small molecules (e.g., AutophagySMDB [87], HAMdb [88]). These resources contain extensive data on autophagy that enables systems-level analyses of its regulation. In the present study, we mined these databases, manually curated the data in the light of recent literature, and identified a structurally and functionally diverse set of 225 drugs that are known to activate, inhibit, or modulate autophagy, as supported by direct experimental evidence. 

The experimentally verified (known) targets of these drugs are available in databases such as DrugBank [25]. However, we are still far from a complete understanding of their actions and side effects. Quantitative systems pharmacology (QSP) methods/tools have been developed with an aim to utilize existing knowledge on networks of interactions among drugs, proteins, and genes to discover new drug targets, reveal underlying mechanisms, and identify repurposed drugs [89,90]. For example, matrix factorization-based machine learning algorithms have been applied to the prediction of novel drug-target interactions [26,27,91]. Pathway/network analysis tools have been developed to study drug effects [28,92,93,94]. 

In this study, our QSP analysis led to the identification of 1837 experimentally verified (known) and 368 computationally predicted (new) interactions between these autophagy modulators and their 993 known and 12 new target proteins, involved in 294 pathways. Many new interactions predicted using our machine learning method through QuartataWeb interface [28] were found to be consistent with recent experiments. For example, an interaction between fluphenazine and dopamine receptor D3 that was not reported in DrugBank (and therefore not included in our input dataset) was predicted with high confidence. Notably, this drug was recently reported to have a subnanomolar binding affinity on DRD3 (*K_i_* = 0.11 nM) [50]. Among targets we predicted, the translocator protein (TSPO) has been reported to inhibit mitophagy downstream of the PTEN-induced kinase 1 (PINK1)-Parkin pathway [95]. Overall, we found experimental evidence in support of 75 (out of 368) predicted pairs, strongly suggesting that the other predictions could as well indicate possible interactions (either repurposable actions or side effects) of existing drugs. Such machine-learning based predictions may thus help generate novel hypotheses worth further testing or interpret experimental observations. As an example, we predict that olanzapine may interact with calmodulin, implying that the reported olanzapine-induced AMPK activation may be partially mediated by the calmodulin-CaMKKβ-AMPK pathway [42,43].

The dominant paradigm in drug discovery based on Ehrlich’s philosophy of magic bullets (drugs selective for single targets) has been challenged by the new concept of magic shotguns (selectively non-selective drugs) [96], due to the recognition of the complexity of diseases such as cancer and neurological disorders and the prevalence of multitarget drugs approved in recent years [97]. In many cases, perturbing individual targets has little effect on disease networks, and thus therapies are required to modulate multiple targets to achieve efficacy. Therefore, though the promiscuity of drugs may lead to unintended side-effects, it could also be an advantage in terms of improving therapeutic efficacy and preventing drug resistance [98]. Several autophagy modulators are found here to be highly promiscuous, including enzyme cofactors copper (145 targets) and zinc (124 targets), and compounds such as fostamatinib (303 targets), artenimol (78 targets), and olanzapine (48 targets). Fostamatinib, the most promiscuous autophagy modulator, is a type I kinase inhibitor targeting the ATP binding pocket of kinases, which is highly conserved across the human kinome. Pharmacological profiling showed that fostamatinib inhibits 120 kinases with IC_50_ values ranging from 3 nM to 3.49 μM [99]. These kinases include crucial ATG proteins such as mTOR, ULK1/2, CaMKKβ, PI3K, AKT, TBK1, PKG, PKA, PDK1, JNK, RAF1, and RTKs (Figure 6a). The atypical antipsychotic olanzapine, on the other hand, binds to receptors HTR2A/2C3/6, DRD1-4, HRH1, ADRA1s, and CHRN1-5 with high affinities (*K_i_* ranges from 4 to 132 nM) at conserved binding sites (e.g., residue C/S3.36) [100]. Olanzapine has been reported to modulate autophagy in both neuronal (e.g., SH-SY5H cells) and non-neuronal cells (e.g., LN229 and T98 glioma cells) by upregulating reactive oxygen species (ROS) and ATG gene expression [44]. 

The dominance of the kinase family members among targets of autophagy modulators directed us to focus more closely on interconnected signal transduction pathways implicated in autophagy modulation (Figure 5). The dominant role of kinases is consistent with a study of the human kinase-substrate post-translational modification network where phosphorylation plays a key role in regulating autophagy [86]. In addition to the kinases highlighted in Figure 5, as illustrated in Figure 6, core autophagy proteins such as ULK1/2 and AMPK are all kinases. MAPKs and PKA crosstalk to other pathways and thus their roles in regulating autophagy depend on the cellular context. Since kinase pathways are activated in several cancers, differential regulation of autophagy by targeting multiple kinase pathways has been suggested for the development of better cancer therapeutics [101]. 

Another important class of proteins broadly used as drug targets, GPCRs, is also targeted for autophagy modulation. In this case, GPCRs regulate second messengers Ca^2+^ and cAMP, and downstream effectors MAPKs, which initiate the major upstream signal transduction that leads to autophagy. It has been suggested that the regulation of autophagy by GPCRs is essential to mediating the effects of hormones/neurotransmitters secreted in response to systemic nutrient fluctuations [58]. Many GPCRs (e.g., ADRA1s) and calcium channels (e.g., CACNA1s) are frequent targets, showing that the corresponding autophagy modulators are mediated by upstream signaling pathways, rather than the canonical mTOR pathway. 

Turning to the NHRs, our analysis highlighted two important transcription factors directedly targeted by autophagy modulators HIF-1α (targeted by PX-478 and 2-methoxyestradiol) and NF-κB (targeted by glucosamine and acetylsalicylic acid). Notably, both of these, and other transcription factors such as FoxO and TFEB, are regulated by estrogen receptor/NHR ESR1, hence the frequent observation of ESR1 as a major target. Another NHR distinguished by its high promiscuity was NR1I2, which directly regulates the kinase pathways.

The key autophagy proteins that these transcription factors and/or NHRs regulate (e.g., Beclin-1, LC3, Atg5) are illustrated in Figure 6a. Though transcription factors such as TFEB, TFE3, and FoxO are not directly targeted by autophagy modulators, their regulators such as AKT, mTOR, and AMPK appear to be targets. The presence of FoxO, HIF-1-, and NF-κB signaling pathways among the top 20 enriched pathways (Figure 4) further confirmed that the regulation of autophagic gene expression is an important mechanism exploited by many autophagy modulators.

Note that Figure 6a only illustrated representative crosstalks between the pathways mentioned above. The signaling network regulating autophagy may be rewired by specific crosstalk in different cell types. For example, in KRAS-driven cancers, calmodulin binds KRAS, and can directly activate PI3K and AKT [102,103]. Further, our classification of autophagy modulators as activators, inhibitors, and dual-modulators is limited by the existing experimental evidence. Though many targets only interact with either activators or inhibitors, given the complex pathway crosstalk shown in Figure 6a, it is possible that current autophagy activators and inhibitors are in fact dual-modulators. For example, the most promiscuous autophagy inhibitor fostamatinib is known to inhibit serine/threonine-protein kinase tousled-like 2 (TLK2), a negative regulator of amino acid starvation-induced autophagy [99]. Thus, fostamatinib may enhance autophagy via TLK2, although there is no experimental data in support of this. Furthermore, Rev-Erb agonists SR-9009 and SR-9011 inhibit autophagy through the nuclear receptor subfamily 1 group D members 1 and 2 (NR1D1/2) [104]. Since NR1D1 mediates the regulation of autophagy rhythms by the circadian clock [104], whether the inhibition of NR1D1 leads to the enhancement or suppression of autophagy might depend on the timing and cellular state. Melanogenesis and TLR signaling pathways are exclusively targeted by autophagy activators or inhibitors, respectively. Whether it implies specific mechanisms underlying autophagy activation and inhibition requires further investigations. 

PKA, downstream of cAMP signaling, takes part in all three tiers of autophagy regulation: (i) it inhibits core autophagy proteins AMPK and mTOR; (ii) it regulates upstream MAPK signaling; and (iii) it inhibits NF-κB signaling. The drugs targeting PKA involve autophagy activators (e.g., fasudil), inhibitors (e.g., fostamatinib), and dual-modulators (e.g., metformin). PKA is a key regulator that governs various cellular processes. Interestingly, a recent high-throughput screening study in yeast suggested that many substrates or downstream autophagy proteins (e.g., Atg10, Atg14, Atg15, and Atg16) regulated by PKA are, in turn, also PKA regulators [105]. This implies the presence of multiple positive and negative feedback loops coupling PKA to autophagic responses. It has been reported that mammalian PKA activates mTORC1 while, in yeast, TORC1 can indirectly activate PKA through a homolog of a ribosomal protein S6 kinase B1 (RPS6KB1) [106]. All these observations point to the utility of a thorough study of the feedback regulation between PKA and Atg proteins in human cells. In this context, dynamical modeling approaches promise to offer helpful pointers [11,107,108,109,110].

As shown in Figure 4, the targets of autophagy modulators are involved in neurological disease, infectious disease, cancer, cardiovascular disease, and metabolic disease pathways; they are upregulated in the liver and brain tissues (Figure 3c). This observation may be partially due to the presence of antipsychotic drugs (e.g., olanzapine and pimozide) and liver disease drugs (e.g., carbamazepine and resveratrol) in our input list of autophagy modulators, both developed for eliminating toxic aggregates. However, it also reflects the crucial role of autophagy in liver and brain health and diseases [111,112,113] and justifies the attempts of repurposing autophagy modulators against liver and brain diseases [21,22,23]. 

The targets involved in cardiovascular diseases are mainly calcium channels. Intracellular Ca^2+^ can both up- and downregulate autophagy and plays a key role in the crosstalk between autophagy and apoptosis [11]. The interplay between autophagy and apoptosis renders the opposing roles of autophagy as a therapeutic target for cardiovascular diseases. For example, in ischemic heart disease, enhanced autophagy is desired for cardioprotective effects via ROS reduction; however, alleviating chemotherapy-induced cardiotoxicity requires suppression of autophagy [114]. 

Apart from cardiovascular diseases, dual-modulation of autophagy is also relevant to designing therapeutic strategies for other diseases such as cancer, infectious diseases, and neurodegenerative diseases. For example, basal autophagy suppresses tumor initiation but promotes cancer progression. During infection, enhancing autophagy facilitates intracellular pathogen clearance but also suppresses immune responses [18]. Thus, precise modulation and time-sensitive strategies are required for autophagy-based therapeutic options. We also note that several drugs proposed for treating neurodegenerative diseases via autophagy are in fact dual-modulators, such as vorinostat, nimodipine, rosiglitazone, sitaglipin, and dexmedetomidine for AD, trifluoperazine and melatonin for Parkinson’s disease (PD), lithium for amyotrophic lateral sclerosis (ALS), berberine for trauma-induced neurodegeneration, ginsenoside Rb for Huntington’s disease, and emodin for hyperhomocysteinemia-induced dementia [115]. Fifty-seven targets of dual-modulators are involved in the calcium signaling pathway and five are microtubule-related proteins: tubulin β and β1 chain (TUBB and TUBB1) and microtubule-associated proteins 2, 4, and τ (MAP2/4/T), which are known to regulate neurodegeneration. For example, mutant MAPT induces disorder and instability in microtubules, which are associated with AD, PD, and ALS [116,117]. Therefore, a deeper understanding of the precise role of dual-modulators in specific disease settings is needed, and caution should be exercised in the use of autophagy dual-modulators for therapeutic purposes.

Our enrichment analysis also shows that many targets of autophagy modulators are involved in apoptosis (e.g., p53, Bcl-2, and caspase 8), necroptosis (e.g., caspase 8, RIPK1, and TNF), and ferroptosis (e.g., p53, 15LOX, and ACSL4) pathways (Figure 4 and Appendix A). In recent work, we have demonstrated the pivotal roles of these targets in regulating distinct cell death pathways [107,118,119,120]. We also found that due to complex crosstalks among pathways, targeting autophagy, necroptosis, and ferroptosis using polypharmacological strategies can lead to better therapeutic effects in terms of cell survival [121,122]. The present study shows that the key regulators of cell death are targeted by autophagy modulators, in accord with the crosstalk between autophagy and these cell death pathways. The integrated network constructed based on our analysis (Figure 6) may serve as an excellent framework for pursuing quantitative analyses shedding light on the biological and biomedical implications of such crosstalks and opening new directions for designing polypharmacological strategies for monitoring or altering cell survival/death decisions. 

## 4. Materials and Methods 

### 4.1. Data Collection

We selected 225 drugs from DrugBank v5.1.1, which have at least one target and are also known as autophagy modulators (Appendix A). We classified the selected modulators into three groups: activators, inhibitors, and dual-modulators by manually searching the literature for direct experimental evidence in support of their specific roles. A dataset of 993 known targets were retrieved from DrugBank (Appendix A).

### 4.2. Drug–Target Interaction Prediction

New drug–target interactions were predicted using our PMF-based machine learning algorithm [26,27,28]. We represented the known interactions between N drugs and M targets as a sparse matrix R. We then decomposed R into a drug matrix U and a target matrix V, by learning the optimal D latent variables to represent each drug and each target using the PMF algorithm. The product of $U^{T}$ and V reconstructed R by assigning each of the unknown entries in R a value, which represents the confidence score for a novel drug–target interaction prediction:
$$R_{N\times M}=U_{N\times D}^{T}V_{D\times M}$$

Using this method, we trained a PMF model based on 14,983 drug–target interactions between 5494 drugs and 2807 targets from DrugBank 5.1.1. We evaluated the confidence scores in the range [0, 1] for each predicted drug–target interaction. We selected the interactions with confidence scores higher than 0.6 within the top 40 predicted targets for each input drug. Performance benchmarking showed that using 0.6 as the threshold confidence score for predictions achieved a good compromise between sensitivity vs. specificity [28]. This led to 368 novel interactions (Appendix A).

### 4.3. Drug–Drug Similarity Analysis

We evaluated the structural similarity between each pair of autophagy modulators by calculating the Tanimoto distance between their 2D structure fingerprints using the RDKit software [123]. We evaluated the functional similarity of two drugs based on their interaction patterns with known targets. Specifically, we represented each drug *i* by a vector $d_{i}$ with values of 0 or 1 (depending on the absence or existence of an interaction between this drug and the corresponding target) in the matrix R. Interaction-pattern similarities between drugs *i* and *j* were then evaluated by calculating the cosine distance between their vectors $d_{i}$ and $d_{j}$:
$$1-\left(d_{i}\cdot d_{j}\right)/\left(\left|d_{i}\left|\;\;\right|d_{j}\right|\right).$$

### 4.4. Pathway Enrichment Analysis

We enriched the KEGG pathways (version March 2018, *homo sapiens*) [124] by the 225 drugs with 993 known and 12 predicted targets. Two hundred ninety-four pathways were identified (see Appendix A). We calculated the hypergeometric p values to rank enriched pathways. Our enrichment scores were defined as follows: Given a list of targets, the enrichment p value for pathway *i* ($P_{i}$) is the probability of randomly drawing $k_{0}$ or more targets that belong to $P_{i}$:
$$P_{i}=\underset{k_{0}\leq k\leq m}{\sum }\frac{\left(\begin{array}{c} K\\ k \end{array}\right)\left(\begin{array}{c} M-K\\ m-k \end{array}\right)}{\left(\begin{array}{c} M\\ m \end{array}\right)}$$ where *m* is the total number of targets we identified, *M* is the total number of human pathways in KEGG, *K* is the number of targets that take part in the examined pathway *i*, and *k_0_* is the number of targets we identified that are involved in pathway *i*. In order to account for multiple testing, we used the Benjamini–Hochberg method [125] to adjust the calculated p values by introducing a false discovery rate (FDR) correction. The FDR is the fraction of false significant pathways maximally expected from the significant pathways we identified, which is bounded by the cutoff of the adjusted p values. Specifically, the adjusted p value, $p_{i}^{*}$, for the *i*th pathway is:
$$p_{i}^{*}=min_{k=i\ldots m}\left\{\min\left(\frac{p_{k}m}{i},1\right)\right\}$$ where *m* is the total number of pathways. Appendix A lists these p values for pathway enrichments based on both known and predicted targets.

## Figures and Tables

**Figure 1 ijms-21-02855-f001:**
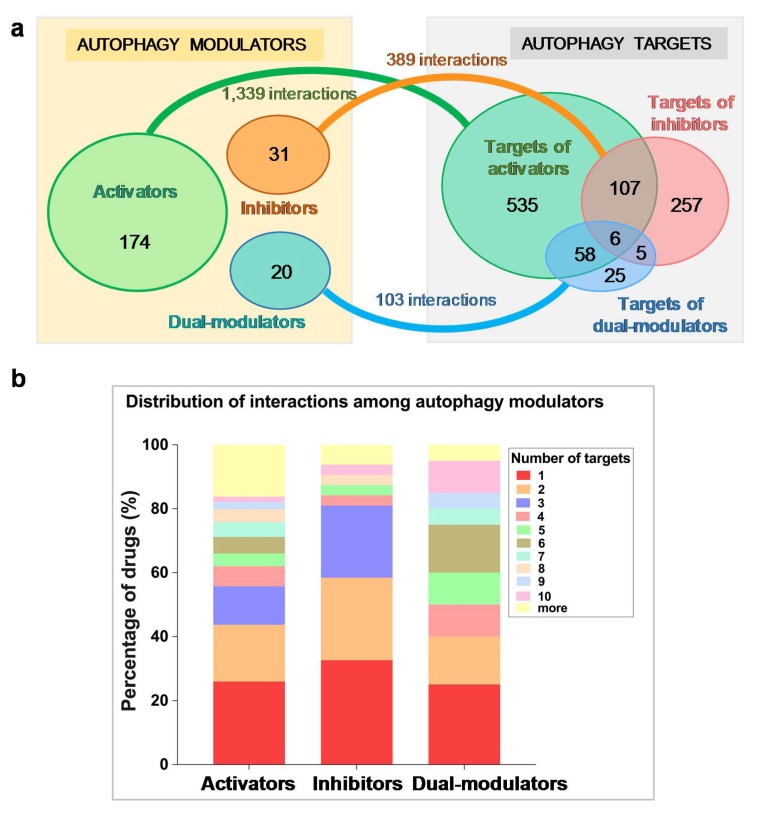
Space of autophagy modulators and their targets. (**a**) On the *left* we show three groups of autophagy modulators, and on the *right* the corresponding targets, which show considerable overlap. The number of targets in each subset are shown by the labels in the Venn diagrams on the *right*. The numbers on the linkers connecting the drugs to the targets indicate the number of targets associated with each category of autophagy modulators; (**b**) Distribution of the degrees (number of associated targets) for each category of autophagy modulators.

**Figure 2 ijms-21-02855-f002:**
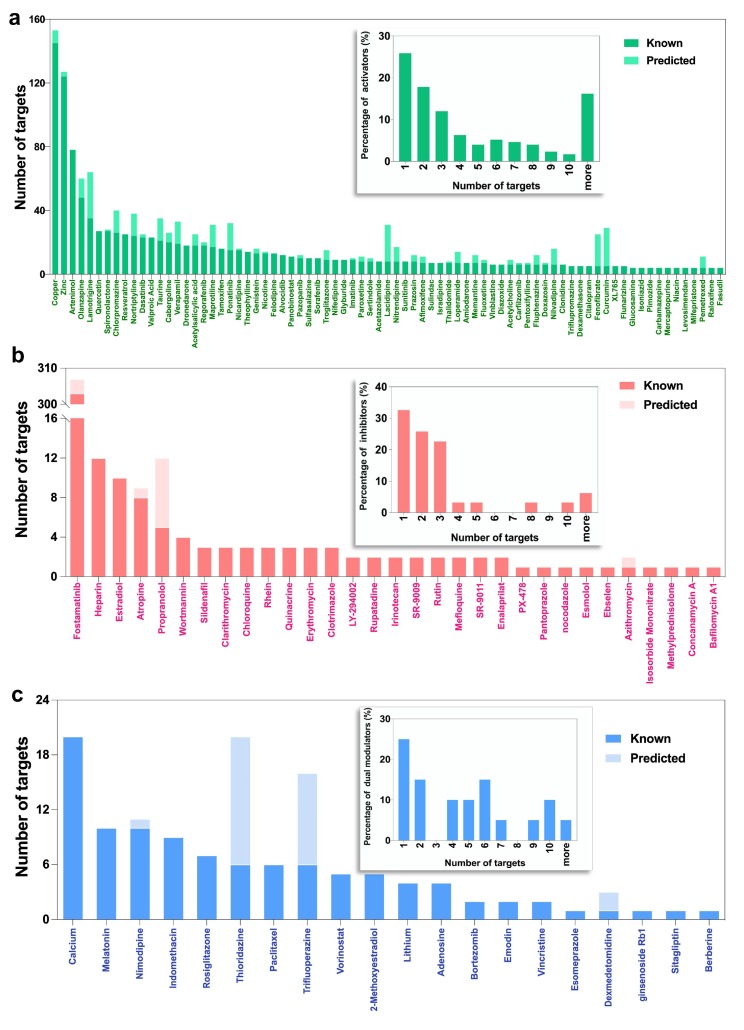
Promiscuity of autophagy modulators. (**a**) The number of known and predicted targets of activators; (**b**) The number of known and predicted targets of inhibitors; (**c**) The number of known and predicted targets of dual-modulators. The drugs are ranked in a descending order of the number of known targets. The contributions of the known and predicted drugs to the total number of targets are shown by the *dark* and *light* colors as indicated by the labels. See also Appendix A for additional data on activators. Insets: Percent distribution of modulators vis-à-vis their promiscuity. The promiscuity is quantified by the degree of each modulator, i.e., the number of targets (*abscissas*) associated with the modulator.

**Figure 3 ijms-21-02855-f003:**
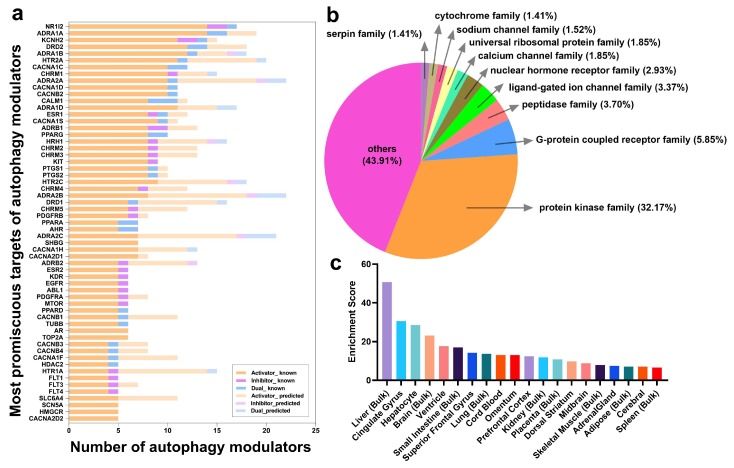
**An overview of proteins targeted by autophagy modulators**. (**a**) Frequent targets of autophagy activators (*orange*), inhibitors (*violet*), and dual-modulators (*blue*), ranked by the number of the total known targets in descending order. In *light* shades are those predicted by our probabilistic matrix factorization (PMF) method. (**b**) Distribution of targeted proteins by protein families; (**c**) Tissues enriched in targets of autophagy modulators.

**Figure 4 ijms-21-02855-f004:**
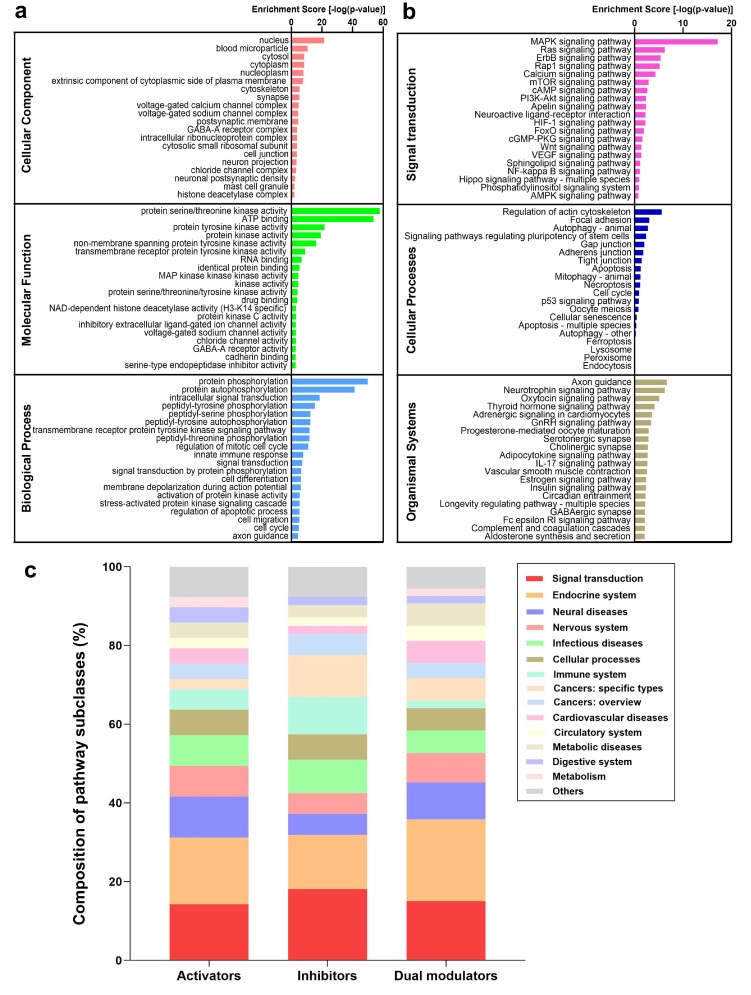
Gene ontology (GO) annotation and pathway enrichment analysis results based on the targets of autophagy modulators. (**a**) Enriched GO annotation terms; (**b**) Enriched KEGG pathways; (**c**) most common pathways in which autophagy targets take part. The distribution of pathway subclasses targeted by autophagy modulators are shown for each category of autophagy modulators. Note the dominance of signal transduction (*red*) and endocrine (*orange*) systems in all categories, the important association of inhibitors with cancers and, to some extent, infectious diseases, and the preferential use of activators and dual modulators in treating neural diseases.

**Figure 5 ijms-21-02855-f005:**
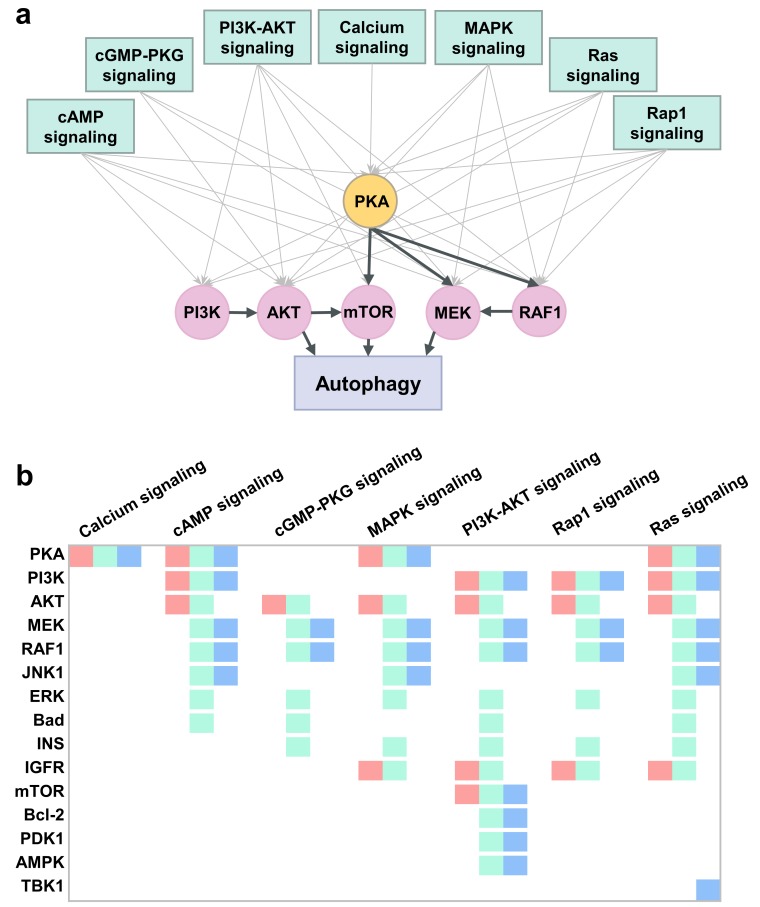
Signal transduction pathways most frequently affected by autophagy modulators and the most promiscuous targets belonging to these pathways. (**a**) Schematic diagram of the seven autophagy-related (ATG)-signal transduction pathways and highly promiscuous targets participating in those pathways (indicated by the *gray* arrow pointing from the pathway to the protein). The target proteins are shown in *light violet* circles, except for a pivotal target (PKA) (shown in *yellow*). Bold face *black* arrows from X to Y means that X regulates Y. (**b**) Target-pathway mapping for the most prominent ATG-signal transduction pathways (*abscissas*) and ATG-signaling targets (*ordinate*). The *color blocks* associated with each protein indicate that it is targeted by autophagy activators (*green*), inhibitors (*red*), and dual-modulators (*blue*). Note that mTOR is also targeted by all three groups of modulators.

**Figure 6 ijms-21-02855-f006:**
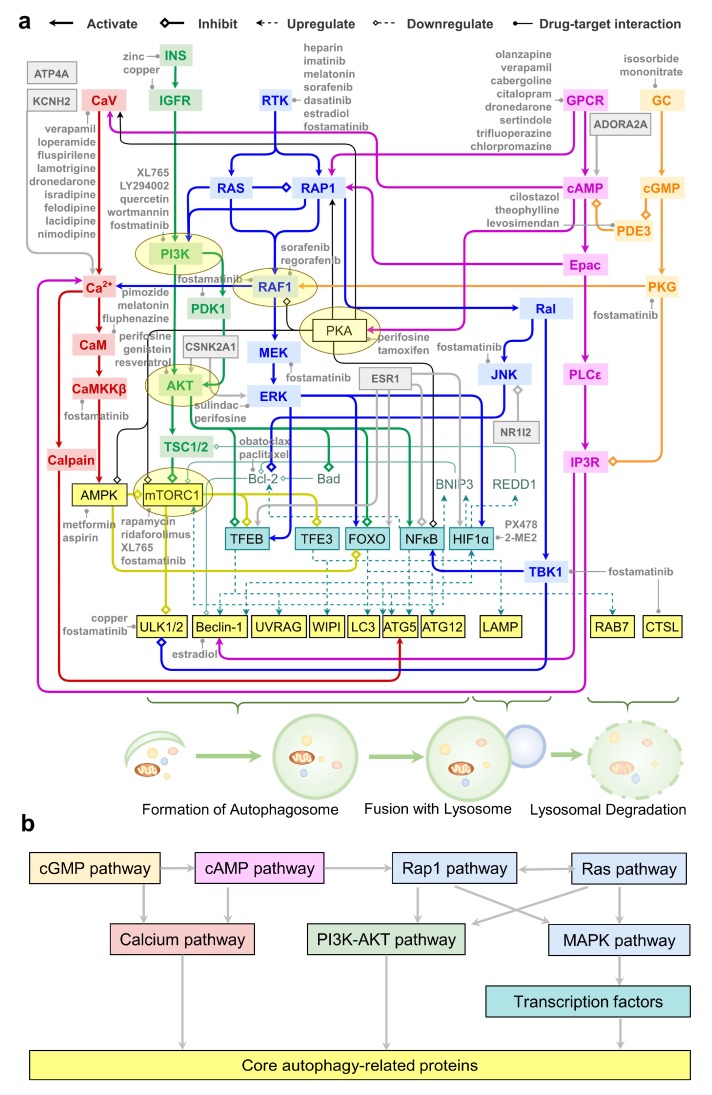
A network of pathways connecting intracellular signal transduction events to autophagy modulation. (**a**) A unified signaling network mediates the effects of autophagy modulators on signaling transduction; (**b**) The regulatory dependency of signaling transduction modules. Color coding: calcium signaling (*red*), cAMP signaling (*violet*), cGMP-PKG signaling (*orange*), Rap1/Ras/MAPK signaling (*blue*), PI3K-AKT signaling (*green*), core autophagy proteins (*yellow*), and transcription factors (*cyan*). The core autophagy proteins were grouped (indicated by braces) by their roles in regulating three phases of autophagy: formation of autophagosome, fusion with lysosome, and lysosomal degradation. The key ATG-signal transduction proteins shown in Figure 5a (except MEK) are highlighted by the *yellow* ellipses. Selected modulators of key targets are listed in *gray* font. See the code for different types of actions indicated by arrow ends on top.

**Table 1 ijms-21-02855-t001:** Experimental validation of predicted drug–target interactions.

Drug	Predicted Target ^1^	Confidence Score	Binding Affinity *K_i_* (nM)	Reference
olanzapine	HTR1F	0.6298	310	[52]
fostamatinib	FGFR4	0.6267	350	[50]
fluphenazine	DRD3	0.6526	0.11	[52]
fluspirilene	DRD3	0.6085	0.40	[52]
thioridazine	DRD3	0.7203	1.5	[52]
sertindole	DRD3	0.8265	2.5	[51]
trifluoperazine	DRD3	0.6003	4.2	[51]
prazosin	ADRA2C	0.8713	10.7	[52]
thioridazine	HRH1	0.6722	16	[52]
chlorpromazine	CHRM5	0.8418	42	[52]
sorafenib	PDGFRA	0.6289	62	[50]
pimozide	HTR1A	0.6044	88	[52]
verapamil	HTR2A	0.6419	140	[52]
fluoxetine	HTR2A	0.7219	148	[52]
maprotiline	DRD1	0.8489	402	[52]
propranolol	HTR2C	0.6593	574	[52]
nortriptyline	HRH2	0.7008	645	[52]
acetylcholine	CHRM5	0.8279	800	[52]
dasatinib	FGFR2	0.6121	1400	[50]
imatinib	FLT3	0.6206	6300	[50]

^1^ Full names of targets are given in Appendix A. See other experimentally validated predictions in Appendix A (rows with binding affinity information in the last column).

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
