# Peer review of "Mechanisms of Action of Autophagy Modulators Dissected by Quantitative Systems Pharmacology Analysis"

_ijms, 2020, doi:10.3390/ijms21082855_

Round 1

Reviewer 1 Report

The manuscript titled “Mechanisms of Action of Autophagy Modulators Dissected by Quantitative Systems Pharmacology Analysis” by Shi et al. aims at evaluating a set of 225 autophagy modulators and carried out a comprehensive quantitative systems pharmacology (QSP) analysis of their targets using both existing databases and predictions made by their machine learning algorithm.

This articles is well designed and discussed.

Author Response

We were pleased to see that the Reviewer was in favor of our manuscript. We thank the Reviewer for carefully reading our manuscript and the nice summary.

Reviewer 2 Report

Minor Considerations:

  1. A) The authors summarize in detail the main findings that have led to in-depth knowledge of the structure and activity of several autophagy modulators. To do this, they focus on a wide range of activators and inhibitors, most of which are widely known and studied in various autophagy models in response to physiological (starvation or hypoxia) and pathophysiological stimuli.

     The authors describe in detail a wide range of different modulators of the autophagy path, for which they use analysis techniques in a coherent and thorough way. They focus the study on various pharmacological modulators, most of which are used in one way or another at the laboratory or even preclinical level in various diseases, both parasitic and of tumor origin.

     In this sense, the review does not provide much information, beyond describing these modulators by action groups. However, the way of presenting them is somewhat tedious for comprehensive reading due to the high number and the multitude of pathways that can be affected.

     A more arduous and focused work on dual modulators of the autophagy path would make much more biomedical sense, considering the relevance of autophagy in diseases such as neurodegenerative diseases, cancer or inflammatory diseases. I recommend to perform a crossover analysis taking into account these same targets during neurodegeneration should be shown.

  1. B) Would it be possible to apply this type of analysis focusing the role of autophagy in development, looking for applicability of modulators in pathologies associated with defects in the functioning of proteins such as Beclin1 or ATG7? For example in the accumulation of fatty tissue in ko ATG7 mice? I would appreciate if the authors replied, no experiments are required, these questions and trying to reason the applicability of their methodology in this type of pathologies.               
  2. 1. C) I recommend improving the quality of the images for easy reading and following the information obtained by the authors. The presentation is adequate but the quality or size of the tables must be improved

Reviewer 3 Report

The authors used a systems pharmacology approach to expand understanding of autophagic pathways by computational analysis of a library of 225 autophagy modulators and known and predicted target proteins. Significantly, this study yielded in silico evidence supporting recent experimental drug-target interaction findings, the observation that existing autophagic modulators tend to affect more frequently regulators of autophagy than core autophagy factors, and highlighting of PKA as a potential central mediator in autophagy-related signal transduction. Additionally, the authors provide several extensive and well-curated datasets that will facilitate subsequent work.   

Discussion: Several modulators including fostamatinib were found to be highly promiscuous in terms of targets. Apart from for copper, zinc and calcium, there is not much discussion regarding why these modulators, especially ones highlighted in the results, are associated with so many targets, or the significance of highly promiscuous modulators. A minor suggestion would be some short general discussion and specific discussion of e.g. fostamatinib and olanzapine on this topic.

Figure 2: The inset graphs seem semi-redundant to supplementary Figure 4.

Table 1: Perhaps include the calculated confidence score in this table.

Figure 3: For panel a, inhibitors are referred to in the figure legend as labeled red; however in the panel they appear violet compared to what is referred to as red in other figures. Please correct either the figure or legend for consistency.

Figure 5: In the figure legend, “pivotal target (PI3K) (shown in yellow)” should refer to PKA rather than PI3K as shown in panel a.

Supplementary Figure S1: References to Fig S1 within the text are somewhat confusing; for example, line 109-110 referring to the “upper right white” region within the “right” map. Labeling each map individually a-f or a/a’, b/b’, c/c’ might clarify.

Supplementary Table S2: Suggest moving note at end of table regarding color coding of certain entries to the top of the table. Suggest adding a similar note at beginning of Table S6.

Line 253-254: Suggest clarifying that the majority of ATG proteins identified in this study as targets of autophagy modulators are not frequent targets. As written, this suggests that the majority of autophagy-related proteins in general are not frequent targets of autophagy modulators. Should similarly be clarified in line 304-305, unless these statements refer to all 137 genes listed in the KEGG pathway cited?

Line 308: Suggest clarifying to “six targets interact with all three categories of modulators.”

Line 610-611: Please elaborate on why 0.6 used as the threshold confidence score.
